# Exploration of the Biocontrol Activity of *Bacillus atrophaeus* Strain HF1 against Pear Valsa Canker Caused by *Valsa pyri*

**DOI:** 10.3390/ijms242015477

**Published:** 2023-10-23

**Authors:** Hongbo Yuan, Bingke Shi, Zhuoni Wang, Genhong Qin, Hui Hou, Hongtao Tu, Li Wang

**Affiliations:** 1Zhengzhou Fruit Research Institute, Chinese Academy of Agricultural Sciences, Zhengzhou 450009, China; yuanhongbo@caas.cn (H.Y.); s18337613180@163.com (B.S.); 18393572039@163.com (Z.W.); 19823988419@163.com (G.Q.); houhui@caas.cn (H.H.); tuhongtao@caas.cn (H.T.); 2Western Agricultural Research Center, Chinese Academy of Agricultural Sciences, Changji 831100, China; 3Zhongyuan Research Center, Chinese Academy of Agricultural Sciences, Xinxiang 453004, China

**Keywords:** pear Valsa canker, *Valsa pyri*, antagonistic biocontrol, *Bacillus atrophaeus*, volatile organic compounds (VOCs)

## Abstract

*Valsa pyri*-induced pear Valsa canker is among the most prevalent diseases to impact pear quality and yields. Biocontrol strategies to control plant disease represent an attractive alternative to the application of fungicides. In this study, the potential utility of *Bacillus atrophaeus* strain HF1 was assessed as a biocontrol agent against pear Valsa canker. Strain HF1 suppressed *V. pyri* mycelium growth by 61.20% and induced the development of malformed hyphae. Both culture filtrate and volatile organic compounds (VOCs) derived from strain HF1 were able to antagonize *V. pyri* growth. Treatment with strain HF1-derived culture filtrate or VOCs also induced the destruction of hyphal cell membranes. Headspace mixtures prepared from strain HF1 were analyzed, leading to the identification of 27 potential VOCs. Of the thirteen pure chemicals tested, iberverin, hexanoic acid, and 2-methylvaleraldehyde exhibited the strongest antifungal effects on *V. pyri*, with respective EC_50_ values of 0.30, 6.65, and 74.07 μL L^−1^. Fumigation treatment of pear twigs with each of these three compounds was also sufficient to prevent the development of pear Valsa canker. As such, these results demonstrate that *B. atrophaeus* strain HF1 and the volatile compounds iberverin, hexanoic acid, and 2-methylvaleraldehyde exhibit promise as novel candidate biocontrol agents against pear Valsa canker.

## 1. Introduction

Pears are among the most popular and important fruits in China, which is the global leader in pear production. The rapid expansion of the pear cultivation area, together with suboptimal cultivation practices and climatic factors, however, have threatened the integrity of pear crops owing to widespread outbreaks of pear Valsa canker caused by *Valsa pyri*, which is most common in northern, northwestern, and northeastern China [1,2]. In some orchards, up to 100% of pear trees are diseased, adversely impacting the sustainable development of the pear industry. Pear Valsa canker primarily impacts the lateral and main branches of pear trees, contributing to bark rot, necrosis, overall tree weakness, and yield losses [3,4]. A series of integrated control strategies are typically employed in an effort to protect against this disease, with fungicides application remaining the most effective and direct approach to controlling pear Valsa canker. However, the widespread and prolonged use of fungicides can impose greater levels of selective pressure on the causative pathogen, in addition to polluting the surrounding environment and negatively affecting food safety, thus limiting the utility of fungicides. There is thus a pressing need for alternative approaches to the reliable prevention and control of this potentially devastating disease.

Endophyte-based biocontrol strategies have emerged as an environmentally friendly alternative to more traditional fungicide use [5]. Several microbes have been shown to effectively facilitate the control of pear Valsa canker to date, such as *Paenibacillus polymyxa* (strain Nl4), *Bacillus velezensis* (strain D4), *Lysobacter enzymogenes* (strain OH11), and *Penicillium citrinum* (strain ZZ1) [6,7,8,9]. While these biocontrol agents exhibit promising applications, just one has been registered in China to date. As such, there is value in the further exploration of other stable and effective antagonistic biocontrol strategies that can help mitigate the spread of pear Valsa canker.

The biocontrol agent *B. atrophaeus* has shown great promise with respect to its ability to suppress the growth of a range of plant pathogens, including *Colletotrichum gloeosporioides*, *Alternaria alternata*, *Fusarium oxysporum*, *C. nymphaeae*, *Rhizoctonia solani*, and *Botryosphaeria dothidea* [10,11,12,13,14,15]. However, few studies to date have focused on the application of *B. atrophaeus* as a biocontrol agent capable of preventing plant diseases caused by *V. pyri*. Mechanistically, *B. atrophaeus* can antagonize pathogen growth by competing for space and nutrients, engaging host defense responses, and/or producing antimicrobial compounds [16]. The production and secretion of bioactive compounds that suppress pathogen growth is the primary antagonistic mechanism utilized by this species. A variety of antagonistic compounds have been characterized to date, including lytic enzymes and peptides that are released into the extracellular environment [16]. *B. atrophaeus* strain CAB-1 has been shown to secrete lipopeptides with robust inhibitory activity against *Botrytis cinerea* [17]. Biocontrol agents are also capable of emitting VOCs that can strongly suppress the growth and survival of plant pathogens [18,19,20]. *B. atrophaeus* strain L193, for example, can produce VOCs that readily suppress mycelial growth for pathogenic species, including *Monilinia fructicola*, *Sclerotinia sclerotiorum*, and *M. laxa*, while also reducing the incidence of *M. laxa*-associated disease [19]. These prior reports thus highlight the promise of *B. atrophaeus*-derived VOCs as candidate biocontrol agents. There is thus clear value in research focused on identifying and characterizing a range of VOCs with antimicrobial activity. To date, researchers have reported several *B. atrophaeus*-derived VOCs with antagonistic activity, including O-anisaldehyde, chloroacetic acid tetradecyl ester, hexadecanoic acid methyl ester, and octadecane [17,18].

Here, *B. atrophaeus* strain HF1 was isolated and found to readily antagonize *V. pyri* growth both in vitro and in vivo. VOCs derived from strain HF1 were further characterized, ultimately leading to the identification of novel VOCs with robust antagonistic activity against *V. pyri*.

## 2. Results

### 2.1. Screening for Antagonistic Endophytic Bacteria

Initial screening efforts led to the isolation and purification of 42 bacterial strains from pear pollen. Dual culture assay results revealed that just thee of these strains were able to effectively inhibit *V. pyri* growth, of which strain HF1 exhibited the strongest antagonistic activity, suppressing mycelial growth by 61.20% (Figure 1A). Microscopic analyses revealed that the hyphae in the control group remained smooth and intact, whereas those that were treated with strain HF1 were thinner with irregular edges (Figure 1B). Together, these results demonstrated that strain HF1 was able to disrupt *V. pyri* mycelial growth and morphological integrity.

### 2.2. Strain HF1 Identification

Based on the biochemical and physiological characteristics observed for strain HF1 (Appendix A), this strain was identified as a member of the genus Bacillus. Phylogenetic trees generated based on the 16S rDNA (accession number: OR673900), gyrA (accession number: OR687234) or groE (accession number: OR687235) sequences from strain HF1 and closely related species indicated that this strain was assigned to the same branch as other *B. atrophaeus* strains (Figure 2A–C). Based on these results, strain HF1 was thus identified as *B. atrophaeus*.

### 2.3. Analyses of the Antifungal Activity of B. atrophaeus Strain HF1 against V. pyri-Induced Pear Valsa Canker

To assess the ability of strain HF1 to inhibit *V. pyri* growth in vivo, strain HF1 cell suspension was next used to treat detached pear twigs that were then inoculated with *V. pyri*. This strain HF1 cell suspension was able to effectively suppress *V. pyri*-induced pear Valsa canker development (Figure 3A). While the average lesion size on CK twigs was 3.93 cm, on strain HF1-treated twigs, this size was reduced to just 2.12 cm (Figure 3B). These findings thus demonstrated the ability of strain HF1 cell suspension to suppress *V. pyri*-induced pear Valsa canker.

### 2.4. The Impact of B. atrophaeus Strain HF1 Culture Filtrate on V. pyri Mycelial Growth and Membrane Permeability

Next, the antifungal activity of culture filtrate prepared from strain HF1 was assessed based on their impact on *V. pyri* mycelial growth. As shown in Figure 4A–C, strain HF1 culture filtrate significantly suppressed *V. pyri* mycelial growth, with inhibition rates of 14.80%, 50.94%, and 64.54% for 2%, 5%, and 10% concentrations of strain HF1 culture filtrate, respectively. Fluorescence observation showed that strain HF1 culture-filtrate–treated hyphae emitted a strong GFP fluorescent signal after SYTOX green staining, while no GFP fluorescent signal was detected in CK hyphae (Figure 5), indicating *V. pyri* hyphae cell membrane permeabilization was affected by strain HF1 culture filtrate. These results thus confirmed that strain HF1 culture filtrate was able to antagonize *V. pyri* growth.

### 2.5. Amplification of B. atrophaeus Strain HF1 Genes Associated with Lipopeptide Biosynthesis

Lipopeptides such as iturin, surfactin, and fengycin are among the most common antimicrobial compounds produced by Bacillus species. To detect whether strain HF1 contained these lipopeptide biosynthesis-related genes, PCR amplification was used. Specific bands of the expected size, as well as *forituD*, *srf*, and *fen* genes, were identified in strain HF1 (Appendix A), and sequencing analyses further confirmed the presence of gene clusters associated with the biosynthesis of iturin, surfactin, and fengycin in strain HF1.

### 2.6. The Impact of B. atrophaeus Strain HF1-Derived VOCs on V. pyri Mycelial Growth and Membrane Permeability

The impact of VOCs derived from strain HF1 on *V. pyri* mycelial growth was evaluated. These analyses demonstrated that these VOCs were able to readily suppress the growth of *V. pyri* mycelia in a dose-dependent fashion (Figure 6A,B). At a strain HF1 concentration of 1 × 10^8^ CFU mL^−1^, these VOCs exhibited an inhibition rate of 84.09% (Figure 6C). These VOCs were also able to readily inhibit a range of other pathogenic fungi that damage fruit plants, with respective mycelial growth inhibition rates for *V. mali*, *B. dothidea*, *C. gloeosporioides*, and *B. cinerea* of 77.90%, 45.52%, 29.2%, and 76.62% (Appendix A). Further analyses demonstrated that strain HF1 VOCs fumigation was sufficient to disrupt hyphal morphology, with treated hyphae appearing curved and thinner when examined via microscopy, whereas CK hyphae were smooth and appropriately shaped (Figure 6D). In addition, *V. pyri* hyphae cell membrane permeabilization was destroyed by strain HF1 VOCs fumigation, since a strong GFP fluorescent signal was detected in strain HF1 VOCs-treated hyphae after SYTOX green staining. There was no GFP fluorescent signal in CK hyphae (Figure 7). These results provided strong evidence that VOCs derived from strain HF1 exhibit high levels of antifungal activity.

### 2.7. Identification of VOCs Produced by B. atrophaeus Strain HF1

An SPME-GC/MS approach was employed to identify VOCs produced by strain HF1. In total, 27 potential VOCs were identified when analyzing 5-day-old strain HF1 samples (excluding the same VOCs in control samples), including alcohols, esters, ketones, alkanes, ethers, acids, and aldehydes (Table 1).

### 2.8. In Vitro Analyses of the Antifungal Activity of Pure VOCs against V. pyri

Thirteen pure compounds including all the Ketones, Esters (except for S-methyl 3-methylbutanethioate, which we could not purchase), Acids and Aldehydes (Table 1) were obtained and tested to gauge their ability to inhibit *V. pyri* growth. These compounds suppressed *V. pyri* to varying degrees at a concentration of 250 mL L^−1^ (Figure 8A,B). Of the tested substances, hexanoic acid, iberverin, and 2-methylvaleraldehyde displayed the highest levels of inhibitory activity such that they were able to fully suppress *V. pyri* mycelial growth (Figure 8A,B). Methyl isobutyl ketone additionally inhibited *V. pyri* growth by 56.85%, while the inhibitory activity of butanoic acid, 2-methyl-, ethyl ester, butanoic acid, 3-methyl-, ethyl ester, methyl methoxyacetate, and methyl isobutyl ketone was relatively weak (35.14%, 30.91%, 27.82%, and 25.14%) (Figure 8B). Hydroxyacetone, 3-methylbutanone, guanidinopropionic acid, bis-(4-fluorophenyl)-methanone, 4-methylphenopentone failed to exert significant antifungal activity against *V. pyri* (Figure 8B). The EC_50_ values for the three strongest antifungal compounds (hexanoic acid, 2-methylvaleraldehyde, and iberverin) against *V. pyri* were 6.65, 74.07, and 0.30 μL L^−1^, respectively (Appendix A).

### 2.9. Analyses of the Antifungal Activity of Pure VOCs against Pear Valsa Canker

Lastly, the antifungal efficacy of purified hexanoic acid, iberverin, and 2-methylvaleraldehyde against pear Valsa canker was assessed using detached pear twigs. This approach revealed that all three of these compounds were able to readily protect against the incidence of pear Valsa canker disease (Figure 9A). Following fumigation with hexanoic acid or iberverin at concentrations of 125 or 250 mL L^−1^, no apparent lesions were observed on pear twigs inoculated with *V. pyri*, whereas on CK twigs, the lesions were 3.24 cm and 3.29 cm on average (Figure 9A,B). In addition, lesions on twigs that had been treated with 125 or 250 μL L^−1^ of 2-methylvaleraldehyde were 1.36 cm and 0.52 cm in size, on average, such that they were smaller than CK treatment (Figure 9B). Together, these results demonstrated that hexanoic acid, iberverin, and 2-methylvaleraldehyde exhibit promise as candidate biocontrol agents that can help to combat pear Valsa canker.

## 3. Discussion

This study was developed with the goal of identifying and characterizing endophytic bacteria with potential utility as biocontrol agents capable of protecting against pear Valsa canker caused by *V. pyri*. Strain HF1 was ultimately identified as the endophyte with strong inhibitor activity. Based on the results of morphological and molecular characterization efforts, this strain was identified as *B. atrophaeus*. *B. atrophaeus* has previously been shown to inhibit the growth of a range of fungal pathogens, including *B. dothidea* and *C. nymphaeae* [10,14]. Consistently, *B. atrophaeus* was herein found to readily antagonize the growth of *V. pyri* in vitro and in vivo in this study. As such, *B. atrophaeus* strain HF1 offers potential value as a novel biocontrol agent that can protect against pear Valsa canker.

*B. atrophaeus* strain HF1 culture filtrate also exhibited significant antagonistic activity, confirming the presence of antimicrobial compounds therein. Lipopeptides are among the main types of antimicrobial compounds produced by *Bacillus* species [21] and include fenygcin, iturin, and surfactin, all of which reportedly exhibit robust antagonistic activity [22]. Zhang et al. [17] found that *B. atrophaeus* strain CAB-1 was able to produce fengycin, while Mu et al. [14] revealed that *B. atrophaeus* strain J-1 harbored genes encoding fengycin, iturin A, and surfactin synthetases. Here, strain HF1 was similarly found to encode genes associated with fengycin, iturin, and surfactin biosynthesis. Endophyte-derived lytic enzymes can directly facilitate pathogen biocontrol. For example, Bacillus and Pseudomonas species can produce proteases and chitinases that can degrade the cell walls of fungal pathogens [23]. Here, strain HF1 secretions were found to exhibit robust protease and chitinase activity (Appendix A), potentially contributing to the observed damage to *V. pyri* mycelial growth and morphological integrity. Moreover, SYTOX green staining confirmed that culture filtrate prepared from strain HF1 induced severe damage to *V. pyri* hyphal membranes. These results clearly demonstrated the ability of strain HF1 culture filtrate to partially mediate the observed antifungal activity of this endophytic bacterial strain through the targeting of fungal cell walls and membranes, in line with antifungal mechanisms reported previously for other biocontrol agents [9,24].

VOCs are secondary metabolites produced by endophytes that can serve as candidate biocontrol agents [25]. Here, strain HF1 was found to exert its antifungal activity against *V. pyri* at least in part through the production of VOCs, consistent with what has been reported for other Bacillus species [18]. Strain HF1-derived VOCs were able to effectively inhibit *V. pyri* mycelial growth while causing pronounced morphological and structural damage of the hyphae, indicating the antimicrobial properties of these VOCs. Further SPME-GC/MS-based characterization led to the identification of 27 specific VOCs present within headspace mixtures prepared for strain HF1. Of these, thirteen pure forms were tested for antimicrobial activity, of which hexanoic acid, iberverin, and 2-methylvaleraldehyde were able to effectively suppress *V. pyri* growth and survival. Although 2-methylvaleraldehyde has previously been identified as a VOC generated by endophytic *Burkholderia pyrrocinia*, they did not directly assess its ability to antagonize pathogenic activity [26]. The sulfur isothiocyanate derivative iberverin presents with strong antioxidant and antitumor activity [27,28], and Lukić et al. [29] previously identified iberverin among Brassicaceae-derived VOCs. There have not yet been any microbe-derived reports of this VOC have yet been reported, however, nor has its antifungal activity against plant pathogens previously been demonstrated. Hexanoic acid has previously been demonstrated to exhibit broad-spectrum antifungal activity against plant pathogens, including *B. cinerea*, *A. solani* and *Xanthomonas citri* subsp. *Citri* [30,31,32]. Following culture in the presence of hexanoic acid, *B. cinerea* mycelial growth, and spore germination were markedly suppressed [31]. Hexanoic acid spraying may additionally activate plant defense responses [30,32]. In the present study, we found that hexanoic acid fumigation also exhibited robust antifungal activity against *V. pyri* growth, with an EC_50_ of 6.65 μL L^−1^. Together, this study is the first to our knowledge to have identified iberverin, hexanoic acid, and 2-methylvaleraldehyde as VOCs capable of readily inhibiting *V. pyri* growth.

Further analyses exploring the antifungal activity of iberverin, hexanoic acid, and 2-methylvaleraldehyde revealed that fumigation with these different purified compounds was sufficient to inhibit pear Valsa canker disease, highlighting their potential utility. Together, these results thus suggest that these three compounds can be effectively leveraged as potential fumigants capable of preventing pear Valsa canker caused by *V. pyri*.

## 4. Materials and Methods

### 4.1. Fungal Pathogens

For this study, the pathogenic *V. pyri* fungal strain lfl-XJ, *B. dothidea* strain Bd220, *C. gloeosporioides* strain Cg467, *B. cinerea* strain Bc1, and *V. mali* strain Vm1 were utilized [33,34]. All of these strains were cultured on potato dextrose agar (PDA; Potato extracts 200 g L^−1^, Glucose 20 g L^−1^, Agar 15 g L^−1^).

### 4.2. Endophytic Bacterial Isolation and Co-Culture Assays

Endophytic bacteria were isolated from pear pollen samples. Briefly, 1 g of pear pollen was added to a 2 mL tube containing 1 mL of ddH_2_O and a steel ball with 0.2 cm diameter. A grinder (SCIENTZ-48, Ningbo city, China) was then used to shake this tube for 1 min (4000 rpm), after which 100 µL of the supernatant fraction was streaked on solid NA medium (Peptone 10 g L^−1^, Beef extract 3.0 g L^−1^, NaCl 5.0 g L^−1^, Agar 15 g L^−1^), which was incubated at 28 °C for 3 days. Any emergent bacterial colonies were then subcultured on fresh plates.

The co-culture of isolated endophytic bacteria and target pathogens was performed as reported previously [9]. At 6 days after co-culture at 25 °C, a Vernier caliper was used to measure the mycelial colony diameter. The control (CK) was just inoculated with target pathogen. The inhibition rate of mycelial growth (%) = (colony diameter of control – colony diameter of co-culture with endophytic bacteria)/(colony diameter of control) × 100%. Assays were performed three times with three replicate samples. An ultra-depth three-dimensional microscope (KEYENCE, Osaka, Japan) was used to examine the impact of antagonistic endophytic microbes on pathogen hyphal morphology following co-culture for 2 days. This experiment was repeated in triplicate, with observations of a minimum of 10 hyphae per experimental replicate.

### 4.3. Identification of Strain HF1

Both morphological and molecular methods were employed for the identification of strain HF1. Morphological identification was performed as in the prior study [35]. Molecular identification was performed by amplifying the strain HF1 16S rDNA, *gyrA*, and *groE* sequences using appropriate primers in Appendix A [36], sequencing the products, and comparing them to the NCBI database to identify homologous sequences. MEGA 7.0 was then used for phylogenetic tree construction using the maximum likelihood method (1000 bootstrap replicates).

### 4.4. In Vivo Analysis of the Antifungal Activity of B. atrophaeus Strain HF1 against V. pyri

The ability of *B. atrophaeus* strain HF1 to control *V. pyri* growth in pear twigs was assessed using protocol from a previously published study [9]. Briefly, healthy 1-year-old pear twigs (Zhongli 1) were cut from a tree, treated for 1 min with 75% ethanol, and each ~10 cm long twig was punched in the center using a 5 mm diameter punch, after which it was sprayed with 1 mL of a strain HF1 cell suspension (1 × 10^8^ CFU mL^−1^) from an overnight culture in liquid NA medium (Peptone 10 g L^−1^, Beef extract 3.0 g L^−1^, NaCl 5.0 g L^−1^) at 180 rpm, 28 °C. Sterile water (CK) and Tebuconazole (0.086 g L^−1^; Anhui Zhongshan Chemical Industry Co., LtD, Anhui, China) treatments were, respectively, used for negative and positive control treatments. Twigs were allowed to dry naturally, after which mycelial plugs (diameter: 5 mm) isolated from the edge of a 5-day-old colony were used for twig inoculation. Mycelial plugs were covered with moistened medical absorbent cotton to maintain local moisture levels, and these inoculated twigs were then transferred into a transparent plastic box for incubation at 25 °C. At 7 days post-inoculation (dpi), Vernier calipers were used to measure lesion length. This experiment was repeated three times, with six inoculation sites per replicate.

### 4.5. Analyses of the Antifungal Activity of B. atrophaeus Strain HF1 Culture Filtrate against V. pyri Mycelial Growth

After culturing strain HF1 for 4 days in liquid NA medium at 28 °C, 180 rpm, cultures were centrifuged and a crude culture filtrate was collected. This filtrate was then passed through a filter (0.22 μm) to yield the final filtrate. PDA medium (~50 °C) was mixed with different concentrations of culture filtrate (2%, 5%, or 10%, *v*/*v*) and used for *V. pyri* plugs (diameter: 5 mm) inoculation. PDA medium without culture filtrate was used as CK. Following a 6-day culture period at 25 °C, colony diameter values were measured. This testing was repeated three times with three replicates per test.

### 4.6. Lipopeptide Biosynthesis-Related Genes Amplification

A DNA extraction kit was used to isolate total DNA from strain HF1, after which the *ituD*, *srf*, and *fen* genes were amplified using appropriate primers in Appendix A [14]. The amplified products were then sequenced by Sangon Biotech Co., Ltd., Shanghai, China.

### 4.7. Analyses of the Antifungal Activity B. atrophaeus Strain HF1-Derived VOCs

A dual Petri dish approach [37] was used to gauge the antifungal activity of VOCs derived from strain HF1 against target plant pathogens. Briefly, 100 µL volumes of strain HF1 cell suspension (1 × 10^6^, 10^7^, or 10^8^ CFU mL^−1^) were evenly coated on an NA plate, with an equal volume of bacteria-free liquid NA medium serving as CK. A mycelial plug (diameter: 5 mm) isolated from the edge of a 5-day-old colony was then inoculated on the center of a PDA plate, and both the inoculated PDA plate and an NA plate were sealed together using Parafilm. At 6 dpi, colony diameters were measured following incubation at 25 °C. These analyses were repeated in triplicate, with three plates per replicate.

### 4.8. Effect of B. atrophaeus Strain HF1 Culture Filtrate or VOCs on V. pyri Hyphae Membrane Permeability

A mycelial plug was put into PDB medium (Potato extracts 200 g L^−1^, Glucose 20 g L^−1^) for inoculation for 2 days at 25 °C, 180 rpm. Mycelium was then collected and processed in strain HF1 culture filtrate. NA liquid-medium-treated mycelium was used as CK. At 24 h after treatment, the mycelium was stained with 1 μM SYTOX green (Biomart, Beijing, China) for GFP fluorescence observation, with the same products as described by Yuan et al. [34].

A *V. pyri* mycelial plug was inoculated on PDA plates covered with cellophane. At 1 dpi, this plate was combined with an NA plate inoculated with 100 µL of strain HF1 cell suspension (1 × 10^8^ CFU mL^−1^), followed by culture for an additional 24 h. Blank NA plate-treated *V. pyri* was used as CK. The hyphae were then stained with 1 μM SYTOX green for GFP fluorescence observation.

### 4.9. Identification of B. atrophaeus Strain HF1-Derived VOCs

In total, 100 µL of strain HF1 liquid culture (1 × 10^8^ CFU mL^−1^) was used to inoculate a headspace bottle containing 5 mL of solid NA medium. Following incubation at 25 °C for 5 days, samples were utilized to identify VOCs through GC/MS in combination with headspace solid phase microextraction (SPME). A headspace bottle that had not been inoculated with strain HF1 served as CK.

Briefly, an SPME fiber (50 µm, DVB/CARon/PDMS) was exposed to bottle headspace for 30 min while incubated at 50 °C, followed by desorption for 10 min in the gas chromatograph injection port at 220 °C while the purge valve was off in split-less mode. Peak separation and detection were performed with an MS detector 5977 MSD (Hewlett-Packard, Geneva, Switzerland) together with a DB-Wax column (30 m × 0.25 mm × 0.25 µm) and the following settings: injection temperature = 260 °C; detection temperature = 220 °C; carrier gas (He) flow rate = 1 mL min^−1^. GC/MS results were analyzed by searching the National Institute of Standards and Technology Mass Spectral Database.

### 4.10. Analysis of the Antifungal Activity of Individual VOCs against Mycelial Growth of Pathogenic Fungi

The ability of the identified VOCs to suppress pathogen growth was assessed as per a previously published method [38]. Thirteen pure standard compounds (Table 1) were obtained for testing their fumigation antagonistic activity against pathogens. A mycelial plug (diameter: 5 mm) from the edge of a 5-day-old colony was placed on a PDA plate that was covered, with a headspace concentration of 250 μL L^−1^ for individual compounds. The mycelial plug inoculation plate without compounds was used as CK. Following incubation for 6 days at 25 °C, colony diameter values were measured. This experiment was repeated in triplicate with three plates per experiment. Concentrations necessary to inhibit mycelial growth by 50% (EC_50_ values) were calculated for each of these compounds using a range of headspace concentrations from 0~250 μL L^−1^, with calculations for pure standards being performed with the DPS software (http://www.dpssoftware.co.uk).

### 4.11. Evaluation of the Antifungal Activity of Specific VOCs against Pear Valsa Canker

This experiment was performed using a slightly modified version of the protocols provided in Section 4.4. Briefly, healthy twigs were injured with a sterile punch, followed by inoculation with mycelial plugs (diameter: 5 mm). Inoculated twigs were then placed into a 1 L plastic box, and the headspace was fumigated with individual VOC standards at headspace concentrations of 125 or 250 μL L^−1^. Boxes were placed in an incubator for 7 days, and lesion length values were then recorded. The inoculated twigs in a box without VOC standards were used as CK. This experiment was repeated three times, with six inoculation sites per replicate.

### 4.12. Statistical Analysis

Significant differences between groups were compared via one-way ANOVAs with Duncan’s multiple range tests using SPSS 20.0 (*p* < 0.01, or 0.05). Data are presented as means ± standard deviations (SDs) from at least three replicates.

## 5. Conclusions

*B. atrophaeus* strain HF1 was identified as a novel biocontrol agent that can protect against pear Valsa canker. Both culture filtrate and VOCs derived from strain HF1 mediated antifungal activity against *V. pyri* growth. Volatile compounds iberverin, hexanoic acid, and 2-methylvaleraldehyde, identified from strain HF1 VOCs, exhibited promise as novel candidate fumigation agents against pear Valsa canker. Overall, our study analyzed the biocontrol activity of *B. atrophaeus* strain HF1 against pear Valsa canker caused by *V. pyri*, laying the foundation for its application.

## Figures and Tables

**Figure 1 ijms-24-15477-f001:**
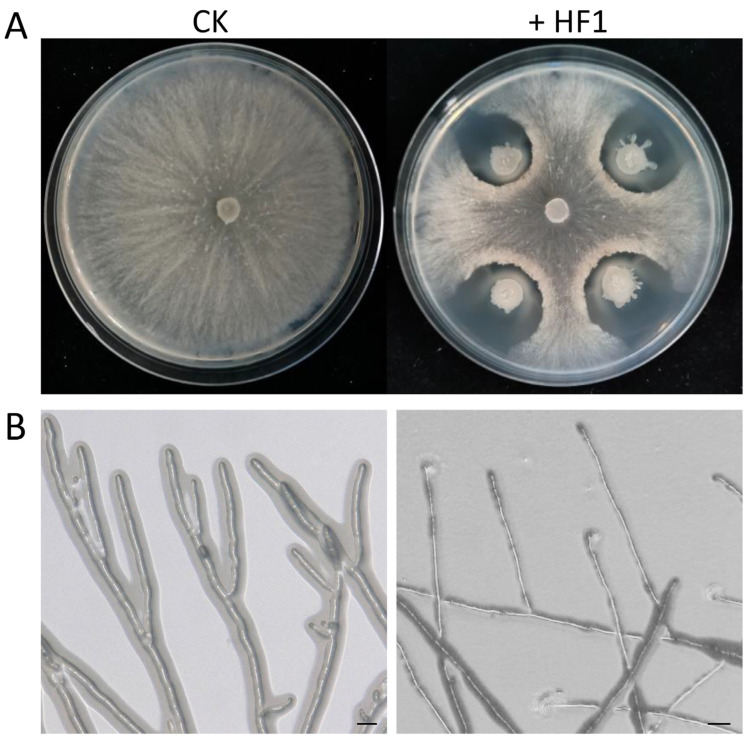
Strain HF1 inhibits *V. pyri* mycelial growth. (**A**) The morphology of *V. pyri* colonies at 6 dpi on PDA medium following exposure to strain HF1. (**B**) *V. pyri* hyphal morphology at 2 days following culture in the presence of strain HF1. Control (CK) *V. pyri* is shown on the left. Strain HF1 treated-*V. pyri* is shown on the right. Scale bar: 20 μm.

**Figure 2 ijms-24-15477-f002:**
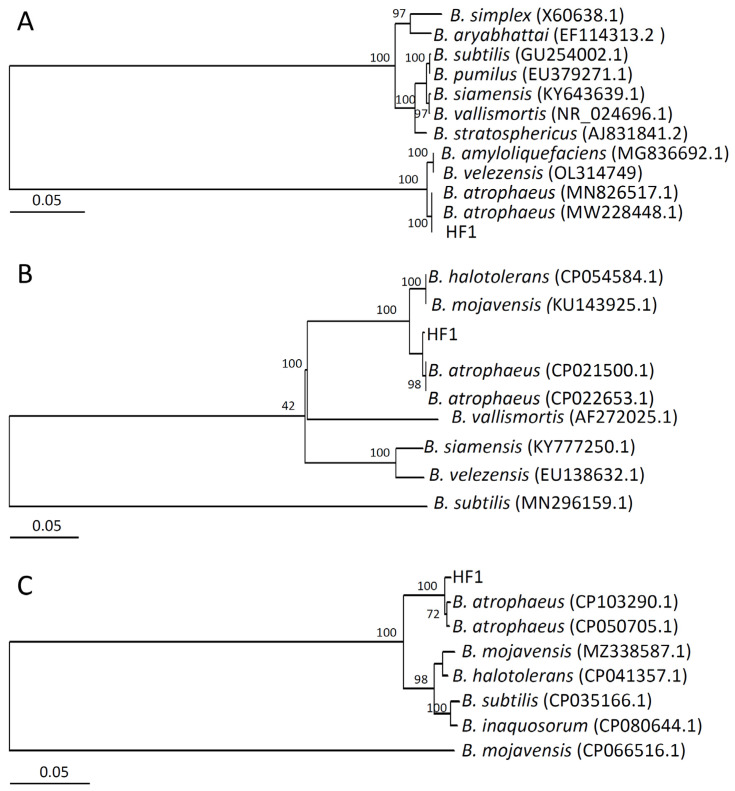
Phylogenetic trees for strain HF1 and related bacterial species. Phylogenetic trees were constructed using 16S rDNA (**A**), gyrA (**B**), and groE (**C**) sequences.

**Figure 3 ijms-24-15477-f003:**
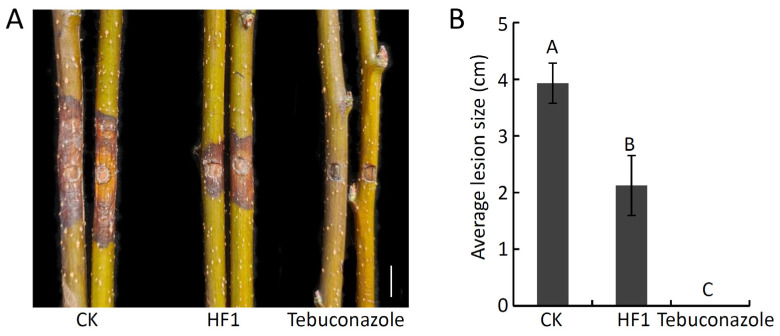
*B. atrophaeus* strain HF1 exerts antifungal activity against pear Valsa canker. (**A**) Biocontrol of pear Valsa canker with strain HF1 cell suspension. Sterile water (CK) and Tebuconazole treatments were, respectively, used for negative and positive control treatments. Scale bar: 1 cm. (**B**) Disease lesion size. Data are means ± SD from six biological replicates. Different letters correspond to significant differences at the *p* < 0.01 threshold. This experiment was independently repeated three times and yielded similar results.

**Figure 4 ijms-24-15477-f004:**
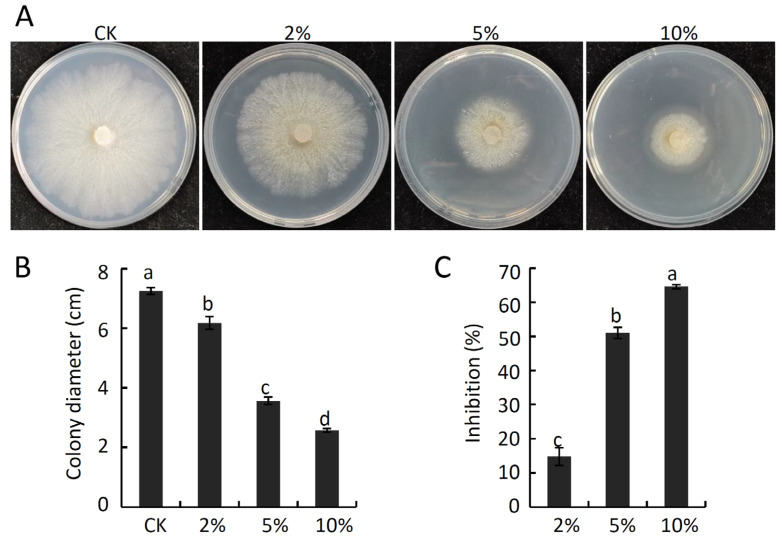
Effect of *B. atrophaeus* strain HF1 culture filtrate on *V. pyri* mycelial growth. (**A**) *V. pyri* colony morphology at 6 dpi on PDA medium containing 2%, 5%, or 10% strain HF1 culture filtrate concentrations. The untreated PDA medium was served as CK. (**B**) *V. pyri* colony diameters. (**C**) Inhibition rates. Data are means ± SD from three biological replicates. Different letters correspond to significant differences at the *p* < 0.05 threshold. This experiment was independently repeated three times and yielded similar results.

**Figure 5 ijms-24-15477-f005:**
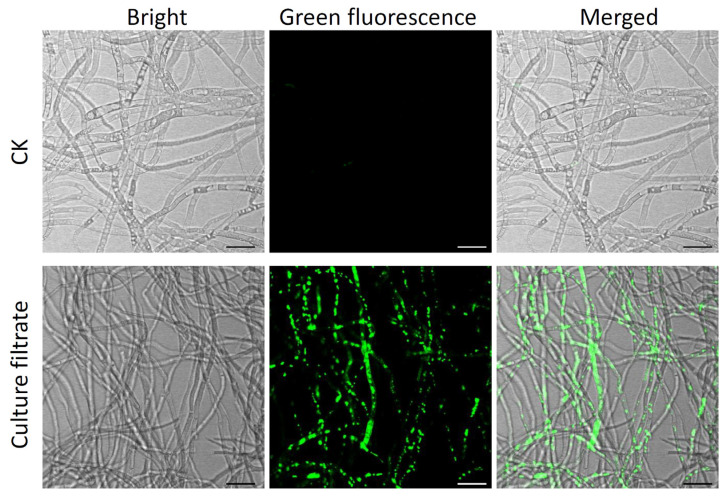
Effect of *B. atrophaeus* strain HF1 culture filtrate on the membrane permeability of *V. pyri* hyphae. The hyphae were stained with 1 μM SYTOX green at 24 h after being treated with strain HF1 culture filtrate. NA liquid medium treatment was used as CK. Bar: 20 μm. Bright: bright-field; green fluorescence: green fluorescence field; merged: merged images of bright-field and green fluorescence-field.

**Figure 6 ijms-24-15477-f006:**
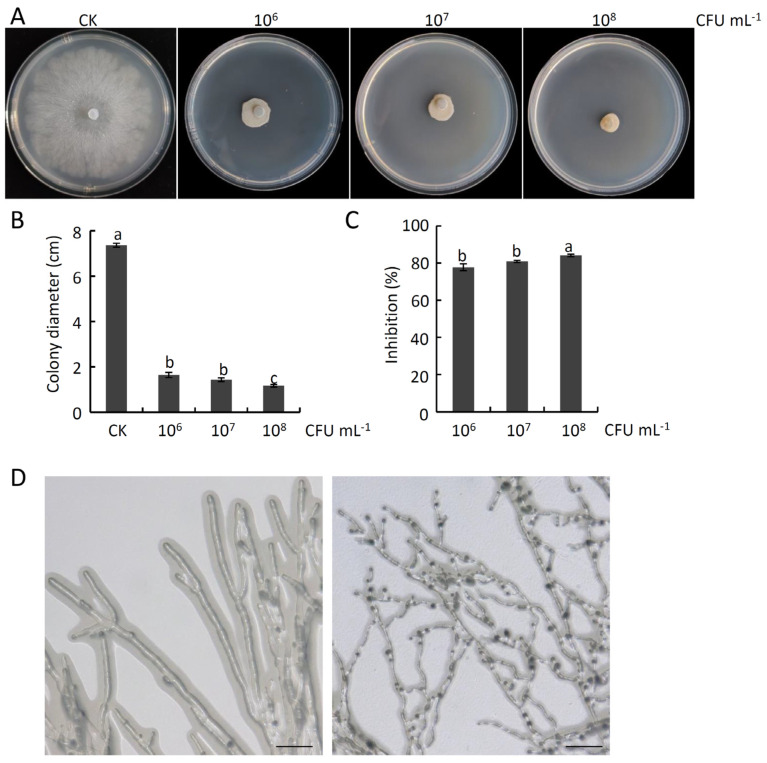
The impact of *B. atrophaeus* strain HF1-derived VOCs on *V. pyri* mycelial growth. (**A**) *V. pyri* colony morphology following fumigation with 100 µL of different concentrations of strain HF1 cell suspension. Bacteria-free liquid NA medium fumigation served as CK. (**B**) *V. pyri* colony diameters. (**C**) Inhibition rates. Data are means ± SD from three biological replicates. Different letters correspond to significant differences at the *p* < 0.05 threshold. This experiment was independently repeated three times and yielded similar results. (**D**) *V. pyri* hyphal morphology. The pictures were taken at two days after fumigation with 100 µL of 1 × 10^8^ CFU mL^−1^ of strain HF1 cell suspension. Left: CK; Right: treatment. Bar: 50 μm.

**Figure 7 ijms-24-15477-f007:**
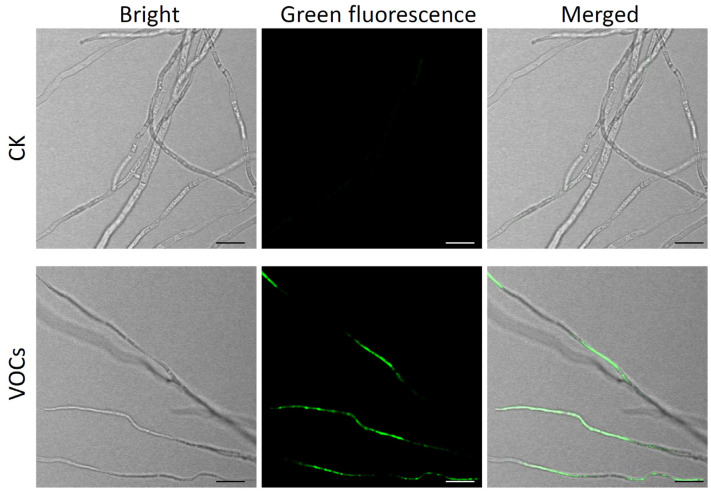
The effect of *B. atrophaeus* strain HF1-derived VOCs on *V. pyri* hyphae membrane permeability. The hyphae were stained with 1 μM SYTOX green at 24 h after fumigation with 100 µL of 1 × 10^8^ CFU mL^−1^ of strain HF1 cell suspension. Bacteria-free liquid NA medium fumigation served as CK. Bar: 20 μm. Bright: bright-field, green fluorescence: green fluorescence-field, merged: merged images of bright-field and green fluorescence-field.

**Figure 8 ijms-24-15477-f008:**
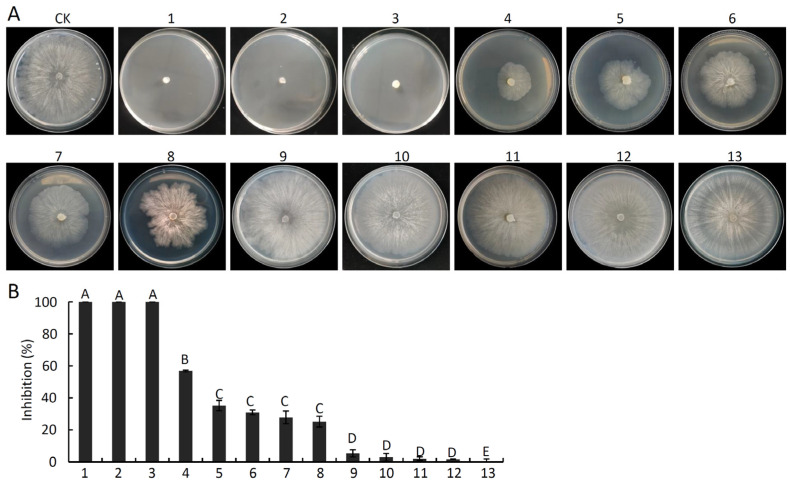
The in vitro antifungal activity of pure compounds against *V. pyri*. (**A**) *V. pyri* colony morphology following fumigation with the indicated pure compounds at a concentration of 250 mL L^−1^. The mycelial plug inoculation plate without fumigation of compounds was used as CK. (**B**) Inhibition rates. 1: Hexanoic acid; 2: 2-methylvaleraldehyde; 3: Iberverin; 4: Methyl thiolacetate; 5: Butanoic acid; 2-methyl-; ethyl ester; 6: Butanoic acid; 3-methyl-; ethyl ester; 7: Methyl methoxyacetate; 8: Methyl isobutyl ketone; 9: Hydroxyacetone; 10: 3-methylbutanone; 11: Guanidinopropionic acid; 12: Bis-(4-fluorophenyl)-methanone; 13: 4-methylphenopentone. Data are means ± SD from three biological replicates. Different letters correspond to significant differences at the *p* < 0.01 threshold. This experiment was independently repeated three times and yielded similar results.

**Figure 9 ijms-24-15477-f009:**
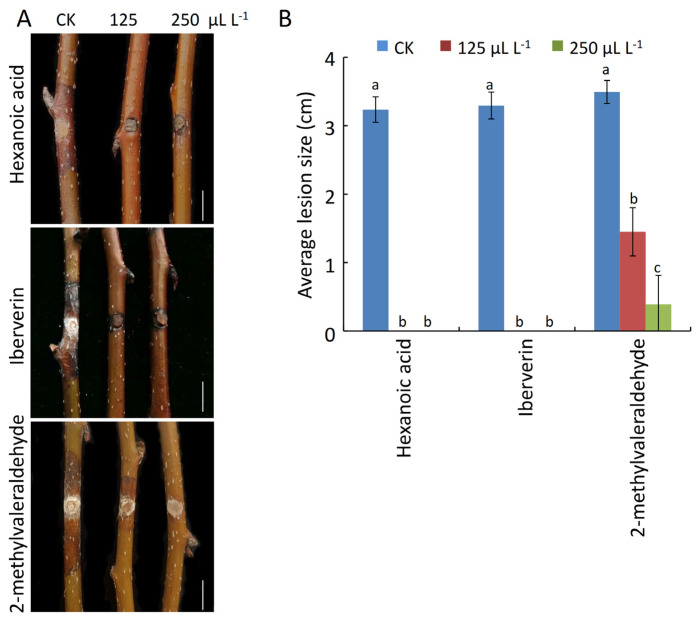
Antifungal activity of pure compounds against *V. pyri*-induced pear Valsa canker. (**A**) Antifungal activity of three pure compounds against *V. pyri* in vivo. Healthy twigs were inoculated with mycelial plugs with following fumigation with the indicated pure compounds (125 or 250 μL L^−1^ headspace). Water was used as CK treatment. Bar: 1 cm. (**B**) Disease lesion size. Data are presented as the mean ± SD of six biological replicates. Different letters represent significantly differences at *p* < 0.05. This experiment was performed independently three times with similar results.

**Table 1 ijms-24-15477-t001:** SPME-GC/MS volatile profile of *B. atrophaeus* strain HF1.

Family	Compound	CAS	Molecular Formula	RT (min)	Area (%)
Alcohols	Methanethiol	74-93-1	CH_4_S	1.87	7.69
	2-pentanol, 3-methyl-	565-60-6	C_6_H_14_O	11.87	8.49
	(±)-5-methyl-2-hexanol	111768-09-3	C_7_H_16_O	14.32	16.56
	2-heptanol, 6-methyl-	4730-22-7	C_8_H_18_O	17.01	16.67
	2-heptanol, 5-methyl-	54630-50-1	C_8_H_18_O	17.39	14.24
	1-hexanol, 4-methyl-	818-49-5	C_7_H_16_O	19.15	5.50
	2-furanmethanol	98-00-0	C_5_H_6_O_2_	24.10	11.15
	Octaethylene glycol	5117-19-1	C_16_H_34_O_9_	40.10	4.03
Esters	Methyl thiolacetate	1534-08-3	C_3_H_6_OS	6.62	19.37
	Butanoic acid, 2-methyl-, ethyl ester	7452-79-1	C_7_H_14_O_2_	6.89	9.99
	Butanoic acid, 3-methyl-, ethyl ester	108-64-5	C_7_H_14_O_2_	7.38	26.80
	S-methyl 3-methylbutanethioate	23747-45-7	C_6_H_12_OS	11.84	8.49
	Iberverin	505-79-3	C_5_H_9_NS_2_	26.35	3.84
	Methyl methoxyacetate	6290-49-9	C_4_H_8_O_3_	29.88	3.75
Ketones	Bis(4-fluorophenyl)methanone	345-92-6	C_13_H_8_F_2_O	2.04	7.21
	3-methyl-2-butanone	563-80-4	C_5_H_10_O	3.72	33.41
	Methyl Isobutyl Ketone	108-10-1	C_6_H_12_O	5.46	6.64
	4′-methylvalerophenone	1671-77-8	C_12_H_16_O	13.48	1.28
	Hydroxyacetone	116-09-6	C_3_H_6_O_2_	14.68	3.95
Alkanes	Octane, 3-ethyl-2,7-dimethyl-	62183-55-5	C_12_H_26_	14.56	3.62
	Decane, 6-ethyl-2-methyl-	62108-21-8	C_13_H_28_	20.23	12.64
	Octane, 2,4,6-trimethyl-	62016-37-9	C_11_H_24_	25.03	7.01
Ethers	1-ethoxy-2-propanol	1569-02-4	C_5_H_12_O_2_	13.09	95.76
	Tri(propylene glycol) methyl ether	25498-49-1	C_10_H_22_O_4_	41.66	5.69
Acids	Guanidinopropionic acid	353-09-3	C_4_H_9_N_3_O_2_	4.02	39.04
	Hexanoic acid	142-62-1	C_6_H_12_O_2_	8.62	5.15
Aldehydes	2-methylvaleraldehyde	123-15-9	C_6_H_12_O	12.33	25.42

## Data Availability

The data analyzed in this study are included within the paper.

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
