# Peer review of "Exploration of the Biocontrol Activity of Bacillus atrophaeus Strain HF1 against Pear Valsa Canker Caused by Valsa pyri"

_ijms, 2023, doi:10.3390/ijms242015477_

Round 1

Reviewer 1 Report

Authors have addressed the control of pear Valsa canker caused by Valsa pyri by using Bacillus atrophaeus strain HF1. Through a series of initial experiments they have shown that the strain significantly growth of V.pyri mycelium. The work also established the importance of various VOCs produced by strains to be effective in the disruption of pathogen’s growth. The major three volatile compounds which showed promising results i.e.  iberverin, 23 hexanoic acid, and 2-methylvaleraldehyde can be used as biocontrol agents for crop management in the future. Overall, the work has been well performed and experiments support the majority of conclusions. However, some figures and results need to be more refined.

Comments

1.    Figure1: better images of the hyphal growth would be better, with more hyphae in the field or some sort of quantification of the phenotype. (i.e., Hyphal length/width ratio)

2.    Label figure 1 to define what left and right panel represent in figure 1A.

3.    Result section 2.5: relevance of amplification of genes should be given to define the purpose of experiment. 

4.    Overall hyphal morphology is not clear in the figures so better resolution would be more informative.

5.    Does the impact of VOC only affect hyphal morphology? In Figure 7 VOC treated hyphae do not look wrinkled at all.

6.    Table 1 formatting is not correct. Lot of overlapping texts.

A minor grammar check is required.

Author Response

Comments

  1. Figure1: better images of the hyphal growth would be better, with more hyphae in the field or some sort of quantification of the phenotype. (i.e., Hyphal length/width ratio)

We have changed the images of Fig 1B.

  1. Label figure 1 to define what left and right panel represent in figure 1A.

We have modified as suggested.

  1. Result section 2.5: relevance of amplification of genes should be given to define the purpose of experiment. 

We added “Lipopeptides such as iturin, surfactin, and fengycin, are among the most common antimicrobial compounds produced by Bacillus species. To detect whether strain HF1 contained these lipopeptide biosynthesis-related genes, PCR amplification was used.” in section 2.5.  

  1. Overall hyphal morphology is not clear in the figures so better resolution would be more informative.

We have replaced Fig 1B and Fig 6D that contained hyphal morphology of V. pyri. In this work, we used the ultra-depth three-dimensional microscope (KEYENCE, Japan) to observe the pathogen hyphal morphology. This microscope can only show the outline morphological characteristics of mycelia. Nest, scanning electron microscopy will be used to observe the effect of strain HF1 on structure of pathogen hyphae.

  1. Does the impact of VOC only affect hyphal morphology? In Figure 7 VOC treated hyphae do not look wrinkled at all.

Strain HF1 VOCs-treated hyphae appeared curved and thinner by comparing with the CK hyphae. We replaced “wrinkled” with “thinner” in Result section 2.6.

  1. Table 1 formatting is not correct. Lot of overlapping texts.

Thank you. We have modified

Reviewer 2 Report

The manuscript is well structured and the objective of the study is well stated.

The Abstract clearly and systematically presents the need of such investigations, the results and input of them. The Introduction is focused on the spread of pear Valsa canker, the strategies for disease control, their advantages and disadvantages, the potential of B. atrophaeus as a biocontrol agent.

The aim of the study is to find endophyte bacteria that are capable to inhibit the growth of V. pyri to be used as a biocontrol agent against pear Valsa canker. Out of 42 bacterial strains from pear pollen were isolated and purified but just 3 of them were able to inhibit effectively the growth of V. pyri. Among them, the HF1 strain exhibited the strongest antagonistic activity, suppressing mycelial growth by 61.20% into in vitro tests and was selected for future analyses. It was identified as B. atrophaeus according to the sequencing data. The results presented in the manuscript show that culture filtrate and VOCs derived from strain HF1 mediated antifungal activity against V. pyri. Experiments were conducted in vitro in a culture media and in vivo with twigs of pear. The ability of three VOCs (iberverin, hexanoic acid and 2-methylvaleraldehyde) to inhibit the growth of V. pyri is proved and reported for the first time.

All results are visualized on nine figures, consisting of photos and graphics, a table and supplementary materials that are informative and evident.

The Discussion made by the authors allows the data presented in the article to be compared with other studies and to evaluate their contribution for development of new approaches for plant protection. The main advantage of this study is the identification of B. atrophaeus strain HF1 as a novel biocontrol agent against pear Valsa canker and his potential for an attractive alternative of fungicides.

All parts into Material and methods section are presented clearly and in detail.

A few technical inaccuracies were found in the text of the manuscript, as follow:

Row 102 – The phylogenetic trees on Figure 2 are too small and it is difficult to read names of bacterial species

Rows 165-167 – the sentence need of correction

Row 220 – “In vitro” is used two times in the title of the section

Row 288 – “proteases” instead of “protases”

Author Response

Row 102 – The phylogenetic trees on Figure 2 are too small and it is difficult to read names of bacterial species.

Thank you. We have modified.

Rows 165-167 – the sentence need of correction

Thank you. The sentence was replaced with “In addition, V. pyri hyphae cell membrane permeabilization was destroyed by strain HF1 VOCs fumigation since strong GFP fluorescent signal was detected in strain HF1 VOCs-treated hyphae after SYTOX green staining. There was no GFP fluorescent signal in CK hyphae (Figure 7).”

Row 220 – “In vitro” is used two times in the title of the section

Thank you. We have modified.

Row 288 – “proteases” instead of “protases”

Thank you. We have modified.

Reviewer 3 Report

The topic of the research is the development of an antagonistic biocontrol against the Valsa canker disease of pears. At the same time, this promised an opportunity for an alternative control, to reduce the environmental damage of chemical control and to produce really quality food.

The planning of the research, the application of the methods and the discussion of the results correspond to the scientifically expected level.

The MS is usually well edited, the text is easy to understand, and moreover, there are almost no typographical errors.

Minor bugs and suggestions:

- ad 26: it is recommended to use antagonistic biocontrol instead of biocontrol,

- ad 101-104: Have you thought about a consensus tree? Maybe it would be better here if it is possible to edit this at all,

- in Fig. In the case of 5., the bright and merged versions were left without explanation,

- ad fig 7. Bar marks are missing from 4 pictures,

- ad table 1.: the editing of the first column slipped,

- ad fig 8. The concentrations used were not specified,

- ad 316: correctly: subsp.,

- ad 332-336: Have all the listed strains been used?

The quality of English in this MS is quite good, just minor editing requiered. 

Author Response

- ad 26: it is recommended to use antagonistic biocontrol instead of biocontrol,

Thank you. We have modified as suggested.

- ad 101-104: Have you thought about a consensus tree? Maybe it would be better here if it is possible to edit this at all,

Thank you for your suggestion. We have tried to construct a phylogenetic tree based on the concatenated 16S rDNA, gyrA and groE sequences, however in same cases we cannot obtain all the three gene sequences in the same strain from NCBI database.

- in Fig. In the case of 5., the bright and merged versions were left without explanation,

Thank you. We have added the explanation in Fig 5 and Fig 7.

- ad fig 7. Bar marks are missing from 4 pictures,

Thank you. We have added the Bar marks.

- ad table 1.: the editing of the first column slipped,

Thank you. We have modified.

- ad fig 8. The concentrations used were not specified,

Thank you. We have added the concentration.

- ad 316: correctly: subsp.,

Thank you. We have modified.